# Secure PD-NOMA with Multi-User Cooperation and User Clustering in Both Uplink and Downlink PD-NOMA

Asif Mahmood [1], Mohamed Marey [1,*], Moustafa M. Nasralla [1], Maged A. Esmail [1] and Hala Mostafa [2]

1   Smart Systems Engineering Laboratory, Department of Communications and Networks Engineering, College of Engineering, Prince Sultan University, P.O. Box 66833, Riyadh 11586, Saudi Arabia; amahmood@psu.edu.sa (A.M.); mnasralla@psu.edu.sa (M.M.N.); mesmail@psu.edu.sa (M.A.E.)
2   Department of Information Technology, College of Computer and Information Sciences, Princess Nourah bint Abdulrahman University, P.O. Box 84428, Riyadh 11671, Saudi Arabia; hfmostafa@pnu.edu.sa
*   Correspondence: mfmmarey@psu.edu.sa

**Abstract:** The power domain non-orthogonal multiple access (PD-NOMA) scheme has gained tremendous interest with the multiple access behavior for fifth-generation (5G) wireless communication. Although the overall performance is improved through accurate power distribution among users' signals, it depends on the user clustering strategy. Moreover, the PD-NOMA communication is not completely secured due to its broadcast nature, which is still a major problem. This paper presents a novel low-complexity short code-based technique utilized by the registered users and the 5G base station (gNodeB) for communication. By doing so, the PD-NOMA scheme is made secure from unregistered users or eavesdroppers. We proposed a three-step user clustering strategy that selects the best cluster among all the possible clusters to improve the overall performance. The proposed clustering strategy achieves a low outage probability in PD-NOMA systems. Moreover, it uses a multi-user decode and forward cooperative relaying scheme with PD-NOMA (Cop-PD-NOMA) to increase the coverage range of the gNodeB. In the multi-user Cop-PD-NOMA, the strong users (near users) are used as relay stations to aid the weak users (far users) by the decode and forward (D&F) technique. The proposed work provides a secure PD-NOMA network and the most effective user clustering approach during validation. The bit-error-rate (BER) comparisons demonstrate that multi-user cooperation outperforms single-user cooperation in Cop-PD-NOMA communication.

**Keywords:** PD-NOMA; multi-user cooperation; user-clustering; secure PD-NOMA

## 1. Introduction

Fifth-generation (5G) applications such as the internet of things, vehicular communication, video conferencing, satellite communication, and wearable devices have come a long way. As a result, wireless communication has grown to include autonomous systems such as self-driving cars, smart homes, advanced cognitive systems, and many more. As a result of this, the number of clients inside the 5G cell has increased, and it has become an ultra-dense cell. In this way, the existing resources, such as orthogonal multiple access (OMA) approaches, which have limits on the number of users due to their orthogonalization, cannot meet the needs of today's wireless communication. To make sure that these needs are met, multiple access schemes were introduced in [1–5], that let different users share the same resources and are capable of serving the congested systems. Among these, non-orthogonal multiple access (NOMA), a promising 5G method [6], is proposed to remove orthogonality restrictions in the frequency and time domains for cells that are closer together.

### 1.1. Existing Literature on PD-NOMA

NOMA techniques are further categorized into power-domain NOMA (PD-NOMA) [7] and code-domain NOMA [8]. In the downlink PD-NOMA, a gNodeB transmits the mul-

tiple users' signals by superimposing them in the power domain. The superposition is performed in a way that lets each NOMA user decode the signal of interest using successive interference cancellation (SIC) at their own receiver [9]. Similarly, in the uplink PD-NOMA, each user sends a signal at maximum power (unless a power control mechanism is used), which is added to the other signals in the air and decoded at the gNodeB using the SIC method [10]. The main difference between uplink and downlink PD-NOMA is the SIC order. Based on the SIC method, each user first separate the high power signal from the composite signal. In the downlink, the near users having better channel conditions first decode and eliminate the far user signal (high powered signal) having poor channel conditions using the SIC method. Once the far user signal is eliminated from the composite signal, the near user then retrieves its signal. While in the uplink, the gNodeB first decodes the signal from the near user and then uses SIC to decode the signal of the far user. The SIC technique is used to cancel the effect of the undesired signal during multi-user detection (MUD) by first decoding it from the combined signal, regenerating and subtracting it from the received combined signal [11]. A novel sky-ground (SG) NOMA receiving structure where the aerial user (AU) and terrestrial users (TU) are communicating with the ground base station is presented in [12]. It is understood that removing the most powerful users will give better performance, but only in the case when the most powerful user's signals are fully correctly decoded at the nearest user, which is impractical. As a result of the decoding errors that occurred during the detection of the far user signal, as well as the complexity of performing SIC operations, the near user experienced performance degradation. The overall performance of the PD-NOMA system can be improved through proper power allocation strategies. The power allocation is completely based on the channel gains of each user [13]. Users with high channel gains (called "strong users") are assigned the least amount of power, while users with low channel gains (called "weak users") are assigned the most amount of power. The two-user PD-NOMA outperforms the NOMA-2000 [14]. In the two user PD-NOMA a single-level of SIC is used by the strong user to eliminate the signal of the weak user. Due to non-orthogonal nature of PD-NOMA the inter users' interference is maximum and it becomes worst if more numbers of users are involved [15,16]. Proper power allocation strategies can help to reduce such interference. In [17], the fixed and dynamic power allocation (FPA and DPA) strategies were proposed for the static and dynamic PD-NOMA environments. The performance of PD-NOMA was analyzed for multi-user scenarios in [18]. The distance-based power allocation is carried out for two, three, and four-users per-cluster PD-NOMA, and the upper limit on the number of users per cluster was proposed in [19].

In ordinary PD-NOMA, the strong users of the system are capable of decoding the weak users' data for SIC operation. In the PD-NOMA system, there are two types of users: (1) Registered users known to the gNodeB via proper communication links. (2) unregistered users or eavesdroppers, who are unknown and can steal or misuse the information from the received composite signal. This way, the privacy of the weak user data is compromised and can be detected by an unintended user. To ensure the data security in PD-NOMA, the physical layer security (PLS) and low probability detection (LPD) methods are highlighted in [20]. Owing to the raised issue, the PLS is proposed in [21,22] to ensure the efficient deployment of PD-NOMA. PLS has been demonstrated to be more effective and reliable than previous upper-layer security methods. This is because it relies on the unpredictability and randomness of the physical layer and uses simple operations to make sure the data being sent is secure. Once the data is secured from the eavesdroppers, the strong users can be used as relay stations that can assist the weak nodes through cooperation. The cooperative relaying PD-NOMA (Cop-PD-NOMA) is presented for the two users system where the strong user (cell-center) is used to decode and forward the signal of the weak user (cell-edge) [23]. By carefully regulating the phases of the passive components at the re-configurable intelligent surface (RIS), a suitable strategy that establishes a re-configurable radio environments of the propagation medium and increases the received signal strength is proposed in [24]. This study has focused on combined beamforming modeling and opti-

mization for RIS-assisted hybrid satellite terrestrial relay systems, in which the connections from the BS and satellite to numerous users were obstructed. Furthermore, in [25,26] rate-splitting multiple access (RSMA) and NOMA are briefly investigated in satellite-terrestrial and aerial integrated networks. The fuzzy logic-based technique for relay node selection was proposed by Taj et al. in [27], which significantly enhances the selection of adaptive data rates and optimal power allocation. To increase system level performance, the cooperative relaying technique uses full duplex and half duplex relaying schemes on demand to improve the performance of cell-edge users by leveraging the cell-center user as a relay node in the PD-NOMA system [28]. For security purposes, the conventional techniques for user authorization in the Internet of Things are ineffective and risky. Conventional biometrics such as face and finger prints are insufficient for security. The authors in [29] suggest finger vein-based user verification to have more highly secure authentication for the Internet of Things. For real-time wireless sensor networks, the guaranteed time slots (GTSs) allocation mechanism is used in IEEE 802.15.4 [30] and an energy-efficient sleep control technique is used in [31], respectively. Moreover, a genetic algorithm based on machine learning was proposed for pedestrian identification in [32].

User clustering or user pairing is an essential factor of the PD-NOMA. Using the clustering technique, it is easy to figure out how much power to give each PD-NOMA user. Three different techniques for sub-optimal user-clustering solutions are provided in [33], all of which improve spectral efficiency while being simple and easy to apply. The maximum energy efficiency can be achieved through the best selections among cell-edge users for user clustering [34]. The likelihood of an outage was lowest in clusters with the most space between users, and it was highest in clusters with the least space between users. In [10], a low-complexity sub-optimal user clustering technique is presented. This technique takes advantage of the fact that different users in a PD-NOMA cluster have different channel gains and splits them up into different clusters to improve the system's overall throughput.

### 1.2. Motivation and Contributions

The majority of research has been focused on power allocation strategies, either fixed or dynamic, which are responsible for overall performance improvement in PD-NOMA. Besides this, some of the literature also addressed the cooperative relaying schemes that are used to aid the weak node users in PD-NOMA systems. Moreover, secure communication in the PD-NOMA system is currently one of the main research areas to be explored. Thus, this study focuses on the major issues that PD-NOMA confronts in both downlink and uplink scenarios and offers solutions to address them. These issues include security, user clustering, and cooperative relaying for performance improvement.

The main contributions of this paper are:

- For secure PD-NOMA communication, this paper presents the spreading-based technique with short codes that ensure the privacy of the PD-NOMA system from unregistered users or eavesdroppers.
- Through multi-user decode and forward (D&F) cooperative relaying schemes in PD-NOMA (Cop-PD-NOMA), strong users (near-users) have been used as relays to help increase the coverage area of the gNodeB and improve the BER performance at weak users (far-users).
- A low-complexity, three-step-based user clustering strategy is proposed to make sure that PD-NOMA clusters have low outage probabilities.

The rest of the article is arranged as follows: In Section 2, the system model for multi-user downlink and uplink PD-NOMA is proposed. Section 3 describes PD-NOMA's proposed secure communication, multi-user cooperation, and user clustering strategy. The simulation results and conclusions are presented in Sections 4 and 5, respectively.

## 2. System Model for Multi-User Downlink and Uplink PD-NOMA

In this section, we explain the basic ideas behind downlink and uplink NOMA for multiple users in a cluster with different channel gains. Multiple signals are superposed

using the different power levels at the transmitter, and the superposed signals are decoded using an SIC mechanism at the receivers.

### 2.1. Multi-User Downlink PD-NOMA

The multi-user downlink PD-NOMA is shown in Figure 1. A gNodeB is used to superpose and transmit multiple distinct signals over the same frequency and time but with different powers non-orthogonally. For simplicity, we consider four users in a single cluster surrounding the gNodeB. All the four users $R_1$, $R_2$, $R_3$, and $R_4$ have distinct channel gains. At each receiver, the desired signal is decoded once all the high-powered signals are eliminated from the superposed signal using the SIC technique. In the downlink PD-NOMA, the signals of strong users are assigned low power, while the signals of weak users are assigned high power. At the gNodeB, all the users' signals are combined, and the composite signal $X$ can be given as follows.

$$X(t) = \sum_{i=1}^{4} \sqrt{P_i} x_i(t), \tag{1}$$

where $P_i$ is the power factor that has been normalized, and $x_i(t)$ is the signal for the $i$th user. The gNodeB total transmit power is denoted by $P_b$, and the sum of the power assigned to all users is less than or equal to the total transmit power, such as,

$$P_1 + P_2 + P_3 + P_4 \leq P_b. \tag{2}$$

Allocation of power is based on the distances and channel gains of each user. Users $R_1$, $R_2$, $R_3$, and $R_4$ have channel gains in the order of $h_1 > h_2 > h_3 > h_4$. Depending on the channel conditions, the power factors are in the range of $P_1 < P_2 < P_3 < P_4$.

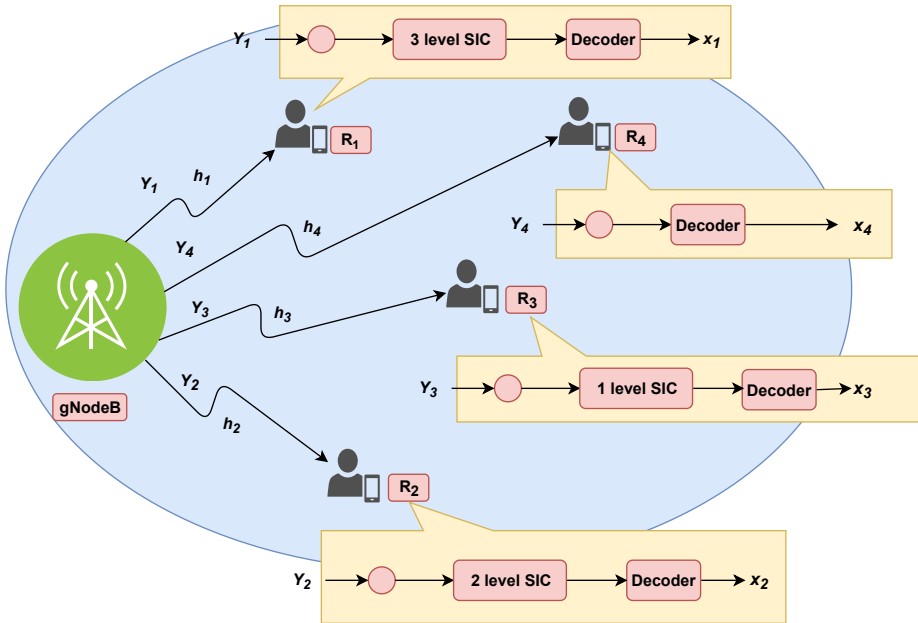

**Figure 1.** Block diagram of multi-user downlink PD-NOMA with the channel gains order $h_1 > h_2 > h_3 > h_4$.

After passing through the channel, the received signal $Y(t)$ for each user is given as follows.

$$Y_i(t) = h_i X(t) + n_i(t), \tag{3}$$

where $h_i$ is the channel gains, which are the reciprocal of the path losses, and $n_i$(t) represents the additive white Gaussian noise (AWGN) for the $i$th user. The Path Loss (PL) is employed in the New York University simulator model (NYUSIM), which is expressed as follows.

$$PL(f,d)[dB] = FSPL(f,1m)[dB] + 10nlog_{10}(d) + AT[dB] + X_\sigma, \qquad (4)$$

where $f$ and $d$ are the frequency in gigahertz and the 3D distance between gNodeB and the $i$th user, respectively. $n$ is the path loss exponent, and $AT$ is the attenuation factor by the atmosphere. $X_\sigma$ is a Gaussian random variable having a mean of zero and a standard deviation of $\sigma$ dB. The $FSPL$ $(f,1m)$ is the free space path loss in decibels at the carrier frequency $f$ and a separation distance of 1 meter. This parameter is defined by,

$$\begin{aligned} FSPL(f,1m)[dB] &= 20log_{10}(\frac{4\pi \times f \times 10^9}{c}), \\ &= 32.4[dB] + 20log_{10}(f), \end{aligned} \qquad (5)$$

where, $c$ denotes the speed of light. To decode user signals, the channel gains ($h_i$) order are used. The highest channel gain user will decode all the users' signals which are lower in channel gains. For the channel gain order, $h_1 > h_2 > h_3 > h_4$: $R_1$ will perform three-level SIC to eliminate the effect of $R_2$, $R_3$, and $R_4$ signals. Similarly, $R_2$ will employ two-level SIC to eliminate $R_3$ and $R_4$ signals, $R_3$ will employ single-level SIC to eliminate $R_4$ signals, and $R_4$ will decode its signal directly due to its low channel gain and high power. In the multi-user downlink scenario, the strong users eliminate the interference caused by the weak users through the SIC technique and hence experience the maximum throughput compared to the weak users. The throughput for user $R_i$ in the multi-user downlink PD-NOMA scenario is given by [35].

$$R_i = log_2\left(1 + \frac{P_i|h_i|^2}{\sum_{j=1}^{i-1} P_j|h_i|^2 + N_i}\right), \qquad \forall \quad i = 1,2,3,4. \qquad (6)$$

where $N_i$ is the noise power spectral density for the $i$th user.

### 2.2. Multi-User Uplink PD-NOMA

In the multi-user uplink PD-NOMA, each user transmits signals at their maximum power of $P_i$ towards the gNodeB. However, each user experiences different channel conditions and hence has different channel gains. A multi-user uplink PD-NOMA system is shown in Figure 2, where four users communicate to gNodeB with the same power and different channel gains. It is understood that PD-NOMA serves users at different power levels, but here all the users have the same transmit power. The users' signals are combined over the air because they are utilizing the same spectrum at the same time. The received composite signal at the gNodeB is given by

$$X = \sum_{i=1}^{4} \sqrt{P_i} h_i x_i(t), \qquad (7)$$

where $P_i$ is the maximum transmit power and $h_i$ is the channel gain for the $i$th user. We know that the channel gain order is: $h_1 > h_2 > h_3 > h_4$, and all the users are transmitting at their maximum power. As a result, the signal of the nearby user $R_1$ with channel gain $h_1$ is attenuated less and has more power remaining in comparison to the signal of the other users. The decoding of the signals at the gNodeB is: $R_1$ signal is decoded first and is used as a SIC term to decode the $R_2$ signal. Similarly, $R_1$ and $R_2$ signals are eliminated using SIC to detect the $R_3$ signal, while $R_1$, $R_2$, and $R_3$ signals are eliminated using SIC to detect the $R_4$ signal.

The achieved data rate in the case of a four-user uplink PD-NOMA is given by [35].

$$R_i = log_2\left(1 + \frac{P_i|h_i|^2}{\sum_{j=i+1}^{4} P_j|h_j|^2 + N_i}\right), \quad \forall \quad i = 1, 2, 3, 4. \tag{8}$$

The main difference between the uplink and the downlink PD-NOMA is the SIC order. In the downlink, the far or weak user data is decoded first, while in the uplink the near or strong user data is decoded first. However, the main theme of the PD-NOMA is preserved in the sense that it decodes the signals based on the power level. In both the uplink and downlink, the signals having the highest power will be decoded first at each receiver. In the downlink, the strong users receive interference-free transmissions upon the successful elimination of the weak user signals through the SIC scheme. While in the uplink, weak users' signals are once decoded upon the elimination of interference from all the strong users.

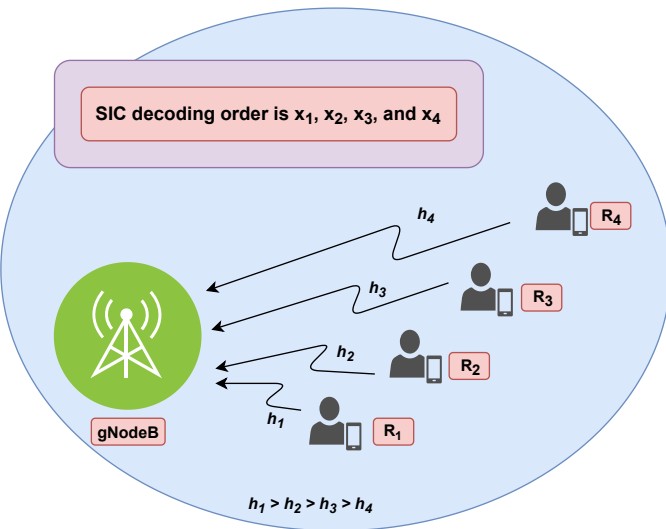

**Figure 2.** Illustration of multi-user Uplink PD-NOMA.

## 3. Proposed Secure Communication, Multi-User Cooperation, and User Clustering Strategy in PD-NOMA

### 3.1. Problem Formulation

This paper focused on three major points in the PD-NOMA systems: (1) Secure PD-NOMA communication; (2) Multi-user cooperation relaying scheme; and (3) User clustering strategy. Due to multiple access and its non-orthogonal nature, NOMA faces more security issues than the other OMA techniques. Due to multi-user detection (MUD) in NOMA, the strong users decode the information of weak users to perform SIC for signal detection. Thus, the security of the NOMA system for the weak or cell-edge users becomes a challenge since it is not designed to conceal the individual's information. The issue is more challenging in the case of densely populated cellular networks. The multi-user PD-NOMA system is shown in Figure 3, where the gNodeB is surrounded by the registered ($R_1, R_2, R_3$, and $R_4$) and unregistered users (eavesdropper, ($E_1$)). Here, the "registered users" are the users of interest to whom the gNodeB wants to communicate. An unregistered user as an eavesdropper is able to steal or misuse the information transmitted from the gNodeB towards the registered users. The eavesdropper is able to decode the information of all the users that are located far from the gNodeB than the eavesdropper. In Figure 3, the downlink scenario for ordinary PD-NOMA is shown where multiple users are served in proximity of the gNodeB. All the users' signals are combined into one composite signal through different power layers and then transmitted. During signal detection, the strong users decode the data of the weak users so that they can do SIC. In traditional PD-NOMA, registered users decode the data of weak users, which is only used for SIC operations.

But the privacy of such systems is compromised if an unregistered user comes close to the gNodeB, which is capable of misusing or stealing the information of remote users. To solve this problem, we came up with a secret-spreading-based method that works with regular PD-NOMA to protect the privacy of these systems. This method is explained in the next Section 3.2. Due to the high frequency of 5G wireless communication, the cell-edge users around the gNodeB are facing severe signal degradation. This signal degradation is less in pico and femto cells but is more in micro and macro cells. To address this issue, it is necessary to use repeaters or relay stations that are capable of aiding these cell-edge users. In this work, we are utilizing the available resources, such as the cell-center users that are used as relay stations and are capable of decoding and forwarding the signals for cell-edge users. This technique is discussed in the Section 3.3. Furthermore, the power allocation in the densely populated 5G cells without user clustering is critical to handle. In the Section 3.4, we offer a three-step low-complexity user clustering technique.

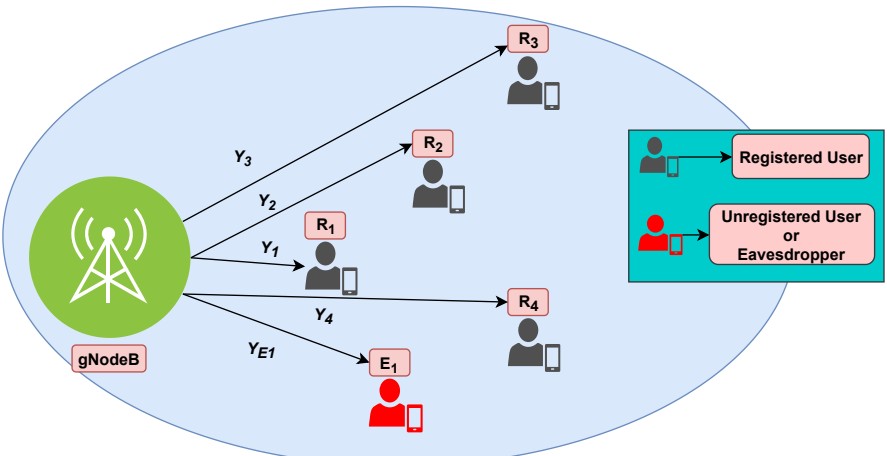

**Figure 3.** Block diagram of downlink PD-NOMA with registered and unregistered users.

*3.2. Proposed Solution for Secure PD-NOMA Communication*

The proposed system shown in Figure 3 assumes a total of five users ($R_1$, $R_2$, $R_3$, $R_4$, and $E_1$). Users $R_1$, $R_2$, $R_3$, and $R_4$ are considered registered users of the system, whereas $E_1$ is an unregistered user. In this case, the strong user is $R_1$, which can decode the data of all the weak users, including $R_2$, $R_3$, and $R_4$. Similarly, $R_2$ can decode both $R_3$ and $R_4$ data, but $R_3$ can only decode $R_4$ data. The proposed technique uses unique short spreading codes that are assigned to gNodeB and registered users. These spreading codes may be orthogonal or non-orthogonal, such as Walsh, pseudorandom sequence (PN), etc. These short codes are used for signal spreading and de-spreading at gNodeB and by registered users. Due to these codes, the registered user and the gNodeB are able to correctly decode the signals from the spread composite signal. The assigned codes may be single-level or multi-level based on the requirements of the system. The proposed system uses single-level codes for securing the users' data from eavesdroppers. However, multi-level codes can also be used to protect complicated systems where eavesdroppers are able to break the security of single-level codes. To validate the proposed technique, we consider the multi-user downlink PD-NOMA scenario where the composite signal at the gNodeB is multiplied with a short secret code and then transmitted over a channel. At the receiving end, each user first de-spread the received composite signal by multiplying it with the same code. The post-processing, such as SIC and demodulation, has been implemented upon successful decoding of the composite signal. The eavesdropper $E_1$, which compromises the ordinary PD-NOMA system security, has no such codes and is not capable of correctly decoding the composite signal. In doing so, the PD-NOMA system's privacy is preserved. In both the uplink and the downlink scenarios, this method works well to keep unintended users from accessing the data. The proposed method helps a lot to protect PD-NOMA communication,

especially in wireless sensor networks (WSN), from unregistered users of the system who might try to break its security.

*3.3. Proposed Solution of Multi-User Cooperation in PD-NOMA*

Cooperative relaying has been receiving a lot of attention in wireless communication because it can provide spatial diversity to minimize fading and get around the problems of putting many antennas on small communication stations. Multiple strong users are used as relay stations and are designated to aid weak users using the D&F technique in cooperative communications. As a result, combining cooperative communications with PD-NOMA can further improve system performance. In this method, users with good channel conditions D&F the signals for other users. This way, weak users with poor channel conditions to the gNodeB can transmit and receive data more reliably.

A downlink cooperative PD-NOMA (Cop-PD-NOMA) is shown in Figure 4, where four users are communicating with the gNodeB. The gNodeB and all the users are equipped with single input and single output (SISO) antennas. Users $R_1$, $R_2$, and $R_3$ are considered strong users, and $R_4$ as a weak user. The $R_4$ signal is attenuated more due to the greater distance separation to the gNodeB. To improve the received signal performance at $R_4$ we use multi-user cooperation from the strong users. In the time slot $(t_1)$, all the users receive a composite signal from the gNodeB, which is given as follows.

$$Y_{i,g}^r(t_1) = h_{i,g} \sum_{i=1}^{4} \sqrt{P_i} X(t_1) + n_{i,g}^r(t_1), \tag{9}$$

where $h_{i,g}$ is the channel gain and $n_{i,g}^r(t_1)$ is the AWGN between the gNodeB and the *i*th user. Superscript *r* denotes signal from gNodeB at *i*th receiver. Each receiver $R_1$, $R_2$, and $R_3$ first decode the $R_4$ signal due to its higher power. The decoded $R_4$ signal is then used for SIC operation to decode the other low powered signals such as $R_3$, $R_2$, and then $R_1$. Users $R_1$, $R_2$, and $R_3$ are used sequentially as relay stations that use the (*D&F*) technique to assist the cell-edge user $R_4$ in the time slots $(t_j)$ where $j = 2, 3,$ and 4. More explicitly, the previously decoded $R_4$ signal for SIC operation is restored and transmitted by each cell-center user. Multiple cell-center users can be used as relay stations, but this means that more bandwidth is being utilized. This way, in the proposed Cop-PD-NOMA system, user $R_4$ receives three forwarded copies of his signal from relay stations. The received D&F signals from each relay station at $R_4$ is given by

$$Y_{4,i}^d(t_j) = l_{4,i} \sqrt{P_4} x_4(t_j) + n_{4,i}^d(t_j), \tag{10}$$

where $l_{4,i}$ denotes the channel gain and $n_{4,i}^r(t_j)$ is the AWGN between $R_4$ and the *i*th user. Superscript *d* denotes the D&F signal from *i*th relay station to $R_4$. Now user $R_4$ have four received signals: one from the gNodeB and three from the relay stations: $R_1$, $R_2$, and $R_3$. $R_4$ uses the maximal ratio combining (MRC) scheme to decode its message signal. The performance improvement due to Cop-PD-NOMA can be observed in the BER curves in the Results section. The generalized equation for D&F Cop-PD-NOMA signal is given by

$$Y_{n,i}^d(t_j) = \sum_{i=1}^{I} \left( l_{n,i} \sqrt{P_n} x_n(t_j) + n_{n,i}^d(t_j) \right), \qquad \forall \quad n = 1, 2...N \tag{11}$$

here, *n* represents the user which receives the *D&F* signals and *i* is the number of relay stations used in the Cop-PD-NOMA system. This way, the Cop-PD-NOMA significantly improve the performance at cell-edge users.

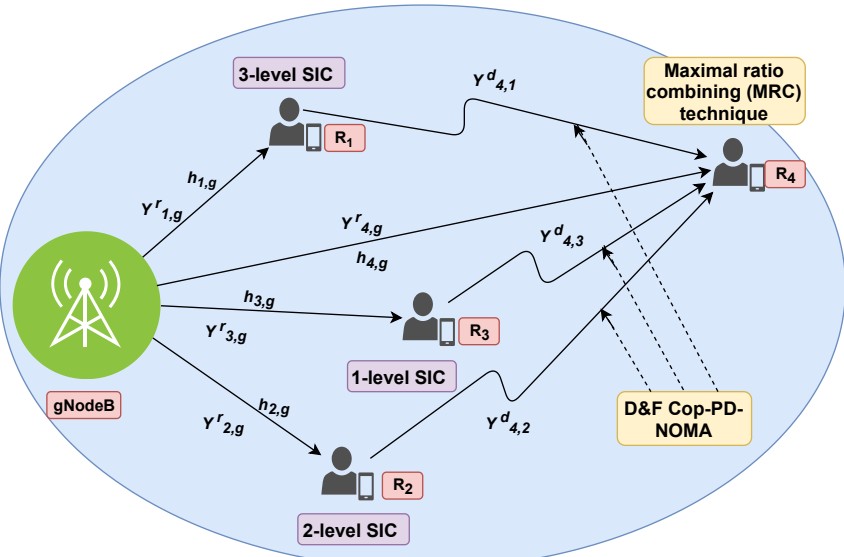

**Figure 4.** Multi-user cooperation in Cop-PD-NOMA with $R_1$, $R_2$, and $R_3$ as strong users and $R_4$ as a weak user.

### 3.4. Proposed Solution for User Clustering Strategy

In the multi-user PD-NOMA system, user clustering or user pairing is as important as the power allocation among the users. The capacity and performance of the PD-NOMA system can be greatly improved by using the proper user clustering strategy. The concept of user grouping is in line with the basic needs of PD-NOMA systems because simultaneous transmissions at the same time and frequency in the denser systems are impractical. The lack of user clustering in the denser PD-NOMA system increases the system overhead for error propagation and channel feedback cooperation. To avoid the system overhead, user clustering can be utilized to improve the capacity and efficiency of PD-NOMA systems. In user clustering, users are divided into multiple groups where PD-NOMA can be effectively employed within each NOMA cluster. Through user clustering, the power distribution among users becomes an easy assignment.

Our proposed user clustering strategy is very helpful in assigning users into different clusters. Suppose we have a total of $N$ users in the vicinity of a gNodeB. The total number of clusters ($C$) equals or less than half of the total number of users ($C \leq N/2$). A minimum of two users and a maximum of $N/2$ users can be assigned to each NOMA cluster. However, an increasing number of users in a cluster results in poor performance. This is due to the multiple levels (m-levels) of SIC that are involved when there are the most users in a cluster. Due to m-level SIC, the complexity of decoding the weak user signals at the strong user increases, which results in poor performance. The best performance can be achieved when users have a maximum distance separation. Our proposed technique uses the following steps to distribute users among clusters:

- Make $C$ clusters with $C \leq N/2$.
- Select a minimum number of users in each cluster to reduce intra-cell interference.
- Each group must contain users with a maximum distance separation.

The proposed three-step technique has been tested on a four-user system and can be expanded to an $N$ number of users distributed across $C$ clusters. All possible cluster arrangements for four users, $R_1$, $R_2$, $R_3$, and $R_4$ are shown in Figure 5. Based on the above three steps, four different cluster types are presented, and each user is supposed to be distance $d$ away from the next user. Using the first step of the proposed algorithm, four different clusters are created. In the Cluster Types 1, 2, and 3, two clusters $C_1$ and $C_2$ are made and assigned with two users each. As shown in Figure 5, Cluster Type 4 has one cluster $C_1$ with four users. The second step of the proposed algorithm is to choose a minimum number of users in each cluster. This is performed to reduce the intra-cell

interference that happens when there are more users within the cluster. For example, Cluster Type 4 has four users and each successive user is distance ($d$) apart from the other. It will use a 3-level SIC technique to decode the signal at near user $R_1$. Similarly, $R_2$ will use a two-level SIC technique, while $R_3$ will use a single-level SIC technique. Due to the increased number of users, the cluster complexity in terms of the multi-level SIC operation and intra-cell interference increases. Hence, Cluster Type 4 fails to achieve the criteria of the second step. On the other hand, Cluster Types 1, 2, and 3 have limited the number of users to a minimum level (each cluster with two users). In these three cluster types, each cluster performs a single-level SIC technique and experiences a minimum intra-cluster interference. Lastly, step three of the proposed algorithm chooses the best cluster based on the rule that each group must have users who are separated by a maximum distance, such as:

$$d(R_i) - d(R_j) > \text{minimum}(d) \qquad \forall \quad i, j = 1, 2 ... N, \quad i \neq j \tag{12}$$

where $d$ is the minimum distance separation between two users. In Cluster Types 2 and 3, the minimum distance separation equals $d$, whereas in Cluster Type 1, it is $2d$. Due to maximum distance separation among the users of a cluster, it achieves the best performance. So, based on steps 1, 2, and 3, Cluster Type 1 is the best possible cluster for the proposed PD-NOMA system. Once the users are distributed among the $C_i$ clusters, the power allocation is then carried out, which is based on the distance separation to the gNodeB as in [19] and given as follows.

$$p'_i = \frac{d(R_i)}{d_{max}}. \tag{13}$$

Here $p'_i$ and $d(R_i)$ represents the $i$th user absolute power factor and the distance separation, respectively. All the absolute power factors are normalized to get the actual power factors ($P_i$) for each cluster, which is defined mathematically by

$$P_i = \frac{p'_i}{\sum_{i=1}^{N/2} p'_i}. \tag{14}$$

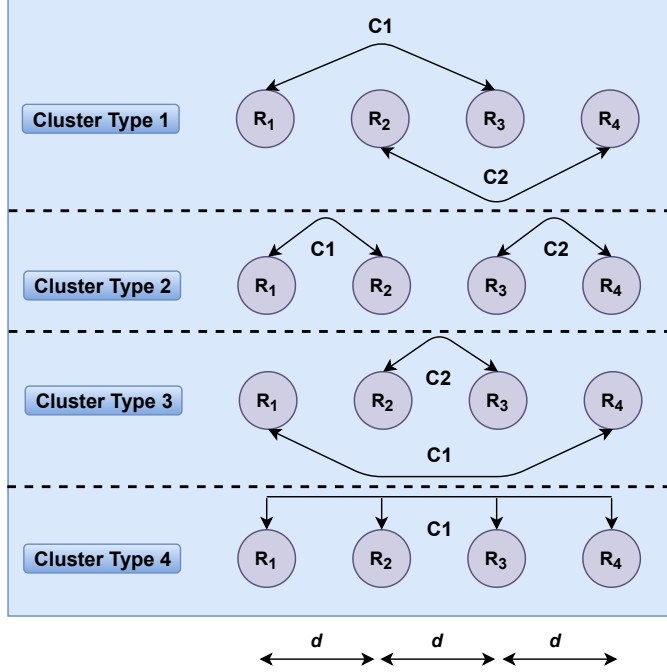

**Figure 5.** Different types of user clustering for four user PD-NOMA ($R_1$ is near and $R_4$ is far to gNodeB).

## 4. Simulation Results

This section provides the performance analysis of the proposed algorithms (secure PD-NOMA, multi-user Cop-PD-NOMA, and user clustering strategy) in both uplink and downlink PD-NOMA systems. Both the downlink and uplink PD-NOMA are tested for a single cluster of four users, $R_1$, $R_2$, $R_3$, and $R_4$. Users' distances are in the order of $d(R_1) < d(R_2) < d(R_3) < d(R_4)$. The performance of the four-user downlink PD-NOMA is depicted in Figure 6, where each user is assigned a power factor that is based on the distance from the gNodeB. Users who are closer to gNodeB are given minimum power, while users who are far away are given maximum power. Figure 6 shows that $R_1$ has a higher BER than other users. This is due to the complexity of performing m-level (3-level) SIC to eliminate the effect of $R_2$, $R_3$, and $R_4$ signals in order to retrieve the $R_1$ signal. BER curves are obtained by averaging repeated simulation runs against different SNR values. Users achieve a threshold BER of $10^{-3}$ at SNR values of: $R_1$ at 19 dB, $R_2$ at 17 dB, $R_3$ at 16.2 dB, and $R_4$ at 14 dB. Similarly, Figure 7 shows the BER performance for the four-user uplink PD-NOMA, because each user transmits at maximum power ($P$) with different channel gains in the uplink PD-NOMA. The received signal power is attenuated more for the users located at a greater distance with lower channel gains than the others. At the gNodeB, $R_1$ signal is decoded first due to its high power, while other users' signals are decoded after the SIC operation to eliminate the high power signals. Figure 7 shows a threshold BER of $10^{-3}$ is achieved at SNR values: $R_1$ at 8 dB, $R_2$ at 16 dB, $R_3$ at 24 dB, and $R_4$ at 29 dB.

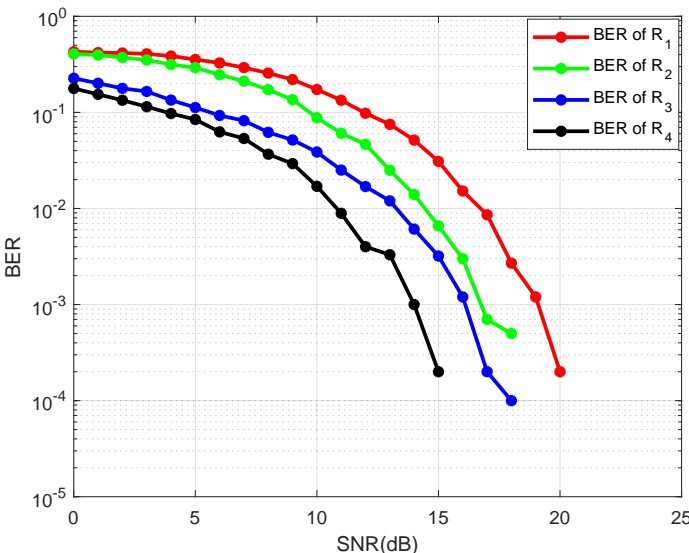

**Figure 6.** BER performance analysis of 4 users downlink PD-NOMA system.

The proposed secure PD-NOMA technique is employed to protect the data from unintended users or eavesdroppers. We used Walsh and PN codes that are only known to gNodeB and registered users, such as $R_1$, $R_2$, $R_3$, and $R_4$. In Figure 3, the downlink scenario is tested for the proposed secure PD-NOMA communication. At the gNodeB, users' signals are combined into a single composite signal and multiplied with their secret code. The spread signal is received by the registered and unregistered users. The registered users ($R_1$, $R_2$, $R_3$, and $R_4$) de-spread the composite signal correctly using their secret code. Post processing, such as SIC, is used to decode and eliminate the high-powered signals. On the other hand, $E_1$ as an unregistered user is not capable of correctly decoding the composite signal due to a lack of the secret code. This way, the data can be protected during the MUD in the multiple access PD-NOMA technique. Figure 8 depicts the BER performance analysis for the proposed technique, where the BER curve for the $E_1$ is a flat line, indicating that the data was incorrectly decoded at the $E_1$ receiver. In the uplink of PD-NOMA, the same

method is used to protect data so that it can be correctly decoded at the desired station, such as the gNodeB.

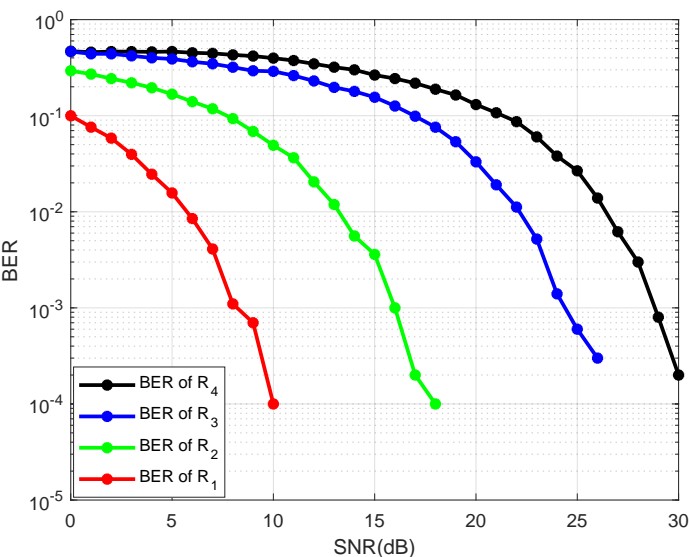

**Figure 7.** BER performance analysis of 4 users uplink PD-NOMA system.

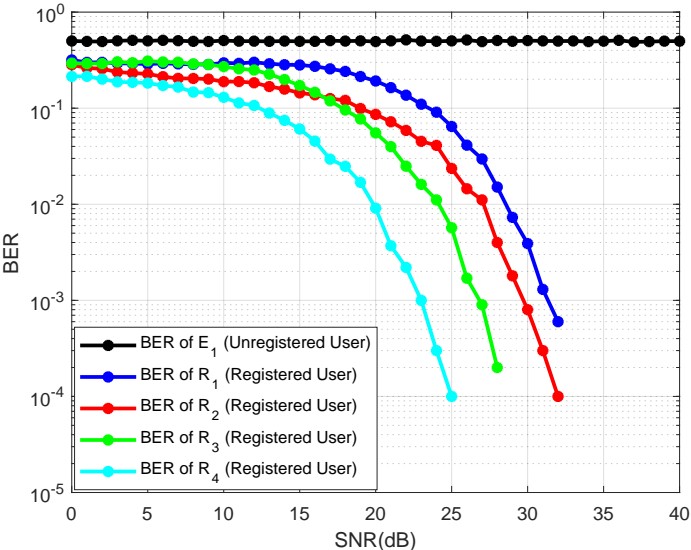

**Figure 8.** BER performance analysis for the registered and unregistered users under the proposed secure PD-NOMA technique.

Presently, we present the performance analysis of the proposed multi-user D&F Cop-PD-NOMA. A total of four users are served around the gNodeB. As shown in Figure 4, the positions of users' $R_1$, $R_2$, and $R_3$ are less than half the distance to the gNodeB as the $R_4$ position. Due to the greater distance, the $R_4$ signal undergoes more attenuation as compared to other users. The BER performance of the proposed system is shown in Figure 9, where three users $R_1$, $R_2$, and $R_3$ act as relay stations, transmitting the D&F signals for the user $R_4$. The $R_4$ performance greatly improves using this multi-user cooperation. The results are compared to single-user cooperation scenarios [23], in which the cell-center user assists the cell-edge user through D&F cooperation. The proposed multi-user D&F Cop-PD-NOMA outperforms the single-user D&F Cop-PD-NOMA. As we can observe in Figures 9 and 10, cooperation by multiple users can improve the BER performance of $R_4$ by 4 dB for the same BER threshold $10^{-3}$, while cooperation by a single user can only improve it by 1 dB or less.

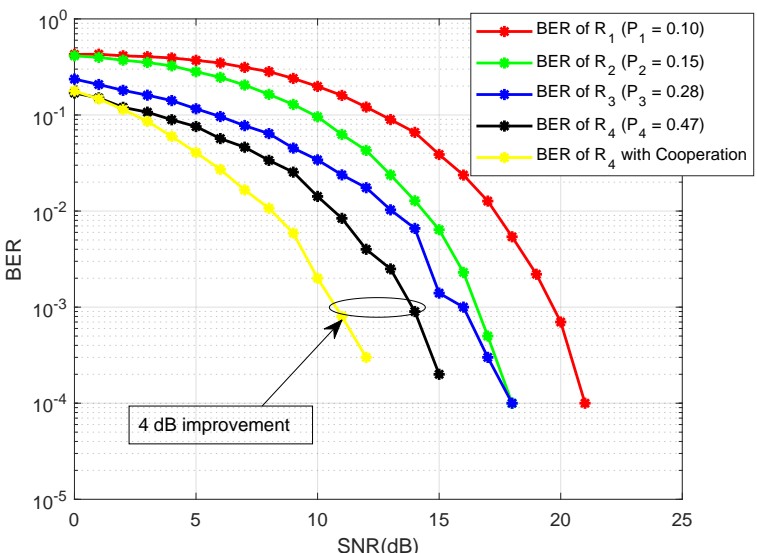

**Figure 9.** BER performance analysis of the proposed four users D&F Cop-PD-NOMA communication.

The performance of the proposed user clustering strategy is tested using outage probability analysis. As discussed in Section 3.4, better performance can be experienced in such clusters that have a maximum distance separation among their users. Two cluster types, cluster A and cluster B, are given with two users, $R_1$ and $R_2$. In cluster A, user $R_1$ has a distance ($d_1$) of 100 m and user $R_2$ has a distance ($d_2$) of 500 m. Similarly, in cluster B, user $R_1$ has $d_1 = 100$ m and user $R_2$ has $d_2 = 150$ m. The distance separation between the users of cluster A is 400 meters, while it is only 50 meters in the case of cluster B. The outage probabilities for both the clusters are plotted vs different SNRs, as shown in Figure 11. Cluster A has a low outage probability because users are as far apart as possible and do not interfere with each other. Cluster B, on the other hand, has a high outage probability because users are close together and interfere with each other at their maximum.

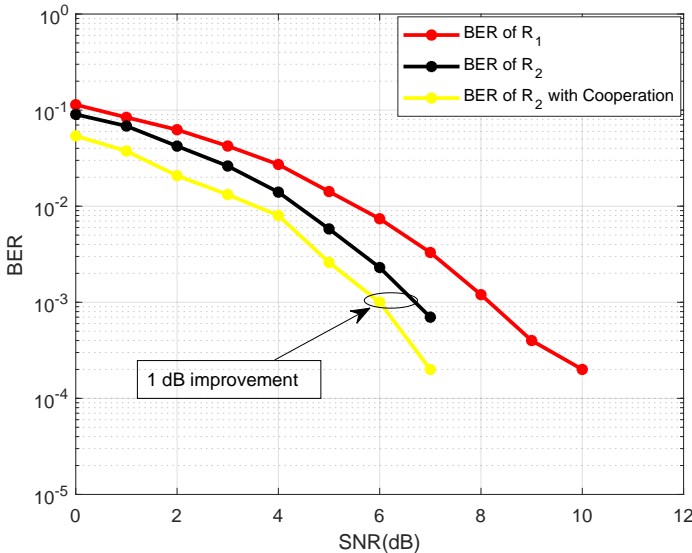

**Figure 10.** BER performance analysis of single-user D&F Cop-PD-NOMA communication [23].

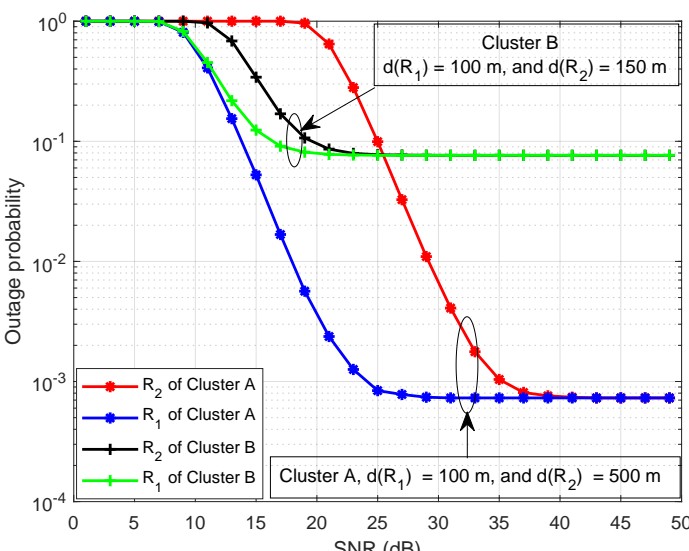

**Figure 11.** Outage probability comparison for different cluster types in PD-NOMA communication.

## 5. Conclusions and the Future Work

The most essential design issues for the PD-NOMA to perform well in both uplink and downlink scenarios are secure communication, cooperation, and power allocation through user clustering. This study describes a spread-based technique using short codes for secure PD-NOMA communication against eavesdroppers who are not registered. The proposed study utilizes the D&F Cop-PD-NOMA technique to improve the performance of weak users. In addition, a low-complexity, three-step user clustering technique is presented to reduce the outage probability that occurs in PD-NOMA systems. The proposed study can be expanded in future work to incorporate more cutting-edge methods that can help to provide secure communication in denser NOMA networks. This strategy can be expanded to accommodate the dynamic PD-NOMA environments' frequency and/or time selectivity. Furthermore, substantial adaptive user clustering can be used to find the best relay stations for cooperation. Additionally, the more advanced PLS techniques will help to determine the optimum solution to the challenges that expose PD-NOMA systems to vulnerability. Overall, the proposed study lays the groundwork for reliable and secure communication in PD-NOMA networks.

**Author Contributions:** Conceptualization, A.M., M.M. and H.M.; formal analysis, M.M. and M.M.N.; data curation, A.M. and M.A.E.; investigation, A.M. and M.M.; methodology, A.M. and H.M.; project administration, M.M.; supervision, M.M., H.M., M.M.N. and M.A.E.; validation, A.M.; writing—original draft, A.M.; writing—review and editing, M.M., M.M.N. and M.A.E. All authors have read and agreed to the published version of the manuscript.

**Funding:** Princess Nourah bint Abdulrahman University Researchers Supporting Project number (PNURSP2022R137), Princess Nourah bint Abdulrahman University, Riyadh, Saudi Arabia.

**Acknowledgments:** The authors would like to acknowledge the support of Prince Sultan University for paying the Article Processing Charges (APC) for this publication.

**Conflicts of Interest:** The authors declare no conflict of interest.

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
