# Peer review of "Secure PD-NOMA with Multi-User Cooperation and User Clustering in Both Uplink and Downlink PD-NOMA"

_electronics, doi:10.3390/electronics11142153_

Round 1
Reviewer 1 Report
1. A novel low-complexity short code-based technique utilized by registered users and 5G base station (gNodeB) is presented for communication. A three-step user clustering strategy that selects the best cluster among all possible clusters in order to improve the overall performance, is proposed. The proposed strategy with the low outage probability is achieved in PD-NOMA systems.
2. Please revise and enhance the English writing to improve the manuscript’s readability.
Reviewer 2 Report
Conclusions must be made better
English language and style must be Improved
Reviewer 3 Report
Regarding the PD-NOMA, the authors present a low-complexity short code-based technique utilized by the registered users and the 5G base station for communication. An user clustering strategy that selects the best cluster among all the possible clusters is also provided.
Look at my criticisms reported below.
- At pag. 2, line 41, you stated "In the downlink, the near user first decodes and eliminates the far user signal using the SIC method": It is not clear to me why. Decoding and removing first the most powerful users shouldn't it give better performance? (See, e.g., https://arxiv.org/pdf/2108.06713.pdf) Please, explain.
- At pag. 3, line 102, you stated "the proposed paper explains in detail how multi-user PD-NOMA works": this is not a novel contribution.
- At pag. 3, line 107, you stated "To aid the performance of the weak users (far users), the strong users (near users) have been used as relays to implement the multi-user decode and forward (D&F) cooperative relaying schemes in PD-NOMA (Cop-PD-NOMA).": How does this affect the overall system capacity (the sum-rate)?.
- In the paper you assume a channel comprising only the path loss; this is not representative into the 5G framework due to great rate demanding and hostile environment. How your approach could be modified to cope with frequency and/or time selectivity? Please explain.
Reviewer 4 Report
Please see the attached file.

Round 2
Reviewer 3 Report
Not all of my suggestions and concerns have been taken into account in the last version of the paper.
Reviewer 4 Report
The authors have well addressed all my concerns, no further comment.
Author Response
Thank you for your positive opinion.
Round 3
Reviewer 3 Report
All of the suggestions have been now implemented.
Author Response
Thank you for your positive feedback about our work.
"All of the suggestions have been now implemented."